# Pre-schoolers' images, intergroup attitudes, and liking of refugee peers in Germany

Iris Würbel[1]*, Patricia Kanngiesser[1,2]

**1** Faculty of Education and Psychology, Freie Universität Berlin, Berlin, Germany, **2** School of Psychology, University of Plymouth, Plymouth, United Kingdom

* iris.wuerbel@fu-berlin.de

## Abstract

There is extensive research on children's intergroup attitudes, but their perceptions of refugee children have rarely been studied. We conducted a study with 5- and 6-year-old children (N = 60) in Germany following the arrival of unprecedented large numbers of refugees in 2015 and 2016. Children completed a set of three tasks that measured their perceptions of refugee children (minority group) and German children (majority group): a draw-a-typical-child task (including questions about whether participants wanted to interact with the depicted child), an intergroup attitude task, and a liking task. Results indicate that participants drew similar pictures of and had similar intentions to interact with refugee children and German children. There was mixed evidence for group favouritism: while participants showed similar explicit attitudes towards German and refugee peers, they indicated more liking of German peers. Moreover, children viewed refugee children as a less variable (more homogeneous) group than German children. Opportunities for intergroup contact with refugee peers (i.e., whether participants attended kindergartens with or without refugee children) had no discernible effect on any of the measures. Our findings provide a snapshot of children's perceptions of refugees in a unique historical context and contribute to research on the development of intergroup attitudes in real-world settings.

## Introduction

Across the globe, people flee from war, persecution, and natural disasters that threaten their life and livelihoods. According to recent estimates, 79.5 million people have been forcibly displaced worldwide (45.7 million people have been displaced in their own country and 33.8 million have fled as refugees or asylum-seekers to other countries), with 40% of them being children [1]. One example of this global refugee emergency is the so-called refugee crisis in Europe since 2015 due to the war in Syria [2]. (Note that we approach the notion of a "refugee crisis" critically as this term often carries a negative connotation [3]). In September 2015, the German government decided to temporarily suspend the Dublin Convention [4], which is a central part of the European Union's asylum system and prescribes that the country where refugees arrive first (or register first) is responsible for processing their asylum claim [5, 6]. At the time, this temporary suspension of the Dublin convention made Germany one of the top five

available upon request from researchers at academic institutions who sign a confidentiality agreement. Please send your data request to the authors or to the Ethics Committee of the Faculty of Education Sciences and Psychology at Freie Universität Berlin: ethikkommission@ewi-psy.fu-berlin.de.

**Funding:** This study was supported by a Freigeist Fellowship from VollswagenStiftung (Volkswagen Foundation) in the form of a grant (Grant No. 89611) awarded to PK.

**Competing interests:** The authors have declared that no competing interests exist.

hosting countries for asylum seekers worldwide [1]. The German public initially showed positive reactions towards refugees and engaged in what has been called "Willkommenskultur" (English: "welcoming culture"): citizens volunteered to help refugees or donated clothing, toiletries, children's toys and other items [7, 8]. However, the following years also saw the rise of voices that described refugees as undeserving "illegitimate migrants" [9, p. 1667], "economic migrants" [5, p. 133] or warned of a "refugee wave" overflowing Germany. Anti-refugee and anti-immigration sentiments were seized on by populist movements such as the political party Alternative for Germany (AfD) and the PEGIDA (Patriotic Europeans Against the Islamization of the West) movement [10, 11]. The latter is often described as a catch-all movement attracting protesters with diverse concerns including right-wing extremist groups and hooligans [12] (in public discourses, hooliganism refers to verbal and physical violence of football fans [13]).

While in 2016 asylum applications peaked at more than 770,000 [14], 14.6% were children younger than six years of age [15]. In the same year, almost 80% of 3- to 6-year-olds with confirmed refugee status attended kindergartens [16]. The specific historical setting of a relatively high number of refugee children entering German early childcare institutions during the so-called refugee crisis in 2015 and the following years, offered a unique opportunity to study the development of intergroup attitudes in real-world settings [17]. Increasingly, researchers have stressed the need to extend studies of intergroup attitudes beyond minimal group settings and to include macro-contextual variables such as historic and geographic factors [18–20]. To date, most research in the context of the so-called refugee crises in Germany has focused on newly arrived refugee children and youth themselves [21–23] and on social relations between and intergroup attitudes of refugee youth and host society youth [24–26]. However, very few studies have focused on young children's perceptions of and attitudes towards refugee peers in this unique historical context in Germany [24].

## Development of social categorization and intergroup attitudes

Social categorization helps individuals to navigate the complexities of their social environment [27]. Social Identity Development Theory describes how children become increasingly involved in and aware of social groups [28]: Children between the ages of two to six years increasingly apply *social categories* to themselves and others, prefer their in-group, and perceive out-groups as homogenous. Specifically, children first categorize others based on gender and then, from age four to five years, based on ethnicity [29–32]. For example, a study with Israeli children found that 5-year-olds relied on ethnicity (Arab/Jew), but not on personality traits or physical appearances, to make inferences about unfamiliar children [30]. Moreover, five-year-old Israeli children used ethnicity more often than other social categories when reasoning about others' preferences [33] and, in contrast to older children, assumed that membership in an ethnic group was permanent [34].

In addition to determining whether someone belongs to a particular group, social categories also influence *perceptions of group variability* [35]. Research with adults has found that out-group members are judged as more homogenous (less variable) than in-group members (out-group homogeneity effect) [36, 37]. In children, a recent study showed that 5- to 8-year-old Israeli and German children viewed out-group members ("Arabs"/"Turks") as more homogenous than in-group members ("Jews"/"Germans") [38]. Moreover, a Portuguese study found that high status children ("white") judged low status children ("black") as more homogenous (out-group homogeneity effect), but low status children viewed their in-group as more homogeneous (in-group homogeneity pattern) [39]. Yet, first observed by Henri Tajfel [40], membership in a "minimal" group based on arbitrary categories, such as blue or red, is

sufficient to induce preferences for the in-group [18, 41]. A study from the U.S. assigned 6- to 9-year-old participants into two colour groups and found that they used favourable extreme ratings ("none" for negative traits and "all" for positive traits) more often for their in-group as compared to the out-group when teachers made daily use of the colour categories [42].

Importantly, processes of social categorization differ from positive or negative evaluations due to (perceived) group membership (affective component of intergroup attitudes) [35, 36, 43], although there seems to be a relation between perceived (out-) group variability and (out-) group prejudice [36]. Developmental research suggests that at the age of five children show substantial in-group favouritism (greater positivity towards their in-group than towards out-groups), while out-group prejudice (more negativity towards out-groups) develops somewhat later from six years of age [44, 45]. A meta-analysis of studies using *explicit* attitude measures, such as attitude trait ratings, found that ethnic, racial, or national biases peaked in middle childhood (5–7 years) and decreased in later childhood (8–10 years) [46]. However, studies using *implicit* measures of intergroup attitudes usually found no decline of bias from middle to late childhood [46–48]. Developmental trajectories for implicit and explicit intergroup attitudes thus appear to differ [49], likely because children increasingly start to control their (explicitly) prejudiced responses [50].

Many studies on the development of intergroup attitudes in children have relied on minimal group paradigms [18, 41, 42]. Recently, scholars have emphasized the need to include macro-contextual variables such as geographic and historic factors into intergroup attitude research with children [18, 19, 51]. To date, these real-world settings have covered a range of groups and contexts. For example, research from Israel has investigated children's intergroup attitudes towards "Jews" and "Arabs" in the context of the enduring "Jewish-Arab conflict" [30, 52, 53]. Researchers in the US have studied children's racial attitudes, focusing on attitudes of "white majority" children (European Americans) towards "black minority" children (African Americans) [54–56]. Furthermore, studies in Canada and Australia have investigated attitudes towards First Nation and Indigenous children [57, 58]. However, there have been few studies on children's attitudes towards refugees.

## Children's attitudes towards refugees

Attitudes towards refugees have been studied extensively in adults in Europe, Australia, Asia, the U.S., and the Middle East [59–64], but there exist only few studies on children and adolescents. According to surveys and interviews, 6- to 11-year-olds in Germany are familiar with refugees and most have met refugee children in school, playgrounds, or during after school activities [24]. Moreover, recent studies have found that adolescents show more inclusive behavioural tendencies towards Syrian refugee peers with good German language skills than to those with bad German skills [26] and that cultural diversity in school settings promotes pro-social intentions towards refugee youth [25]. Relatedly, a study with 9- to 12-year-olds in the Netherlands showed that perceived similarity between Dutch and refugee children correlated with positive intergroup attitudes towards refugee children [65]. While these studies have provided insights into older children's and adolescents attitudes, there is a lack of studies on young children's (e.g., 5- to 6-year-olds') attitudes towards refugees, particularly in the unique historical context of the so-called refugee crises in Germany.

## Intergroup contact

Contact with different out-groups or minority groups can have a positive effect on intergroup attitudes [66–69] and even mere exposure can positively influence attitudes [70, 71]. Programmes that facilitate opportunities for direct interethnic contact have been shown to be

effective even in regions with severe (current or historic) intergroup conflicts, such as the Middle East (Jewish and Arab-Palestinian persons) and Northern Ireland/Republic of Ireland [72]. The benefits of direct intergroup contact have also been demonstrated in educational settings [73, 74]. For example, a study in the US found that European American children in homogenous schools (with limited contact opportunities) had a greater out-group bias towards African American children than children from more heterogeneous schools [75]. A meta-analysis revealed that structured child and youth programmes with direct contact improved young people's intergroup attitudes more than those with indirect contact [76].

Some studies have investigated the effects of contact interventions on attitudes towards refugee children. Cameron et al. [77] studied different extended contact interventions by presenting English 5- to 11-year-old children with different friendship stories including English and refugee children. Their interventions had an overall positive effect on children's attitudes and their intended behaviours towards refugees. A more recent study with 8- to 9-year-olds in Turkey showed some limited impact of an indirect contact intervention (reading stories on positive intergroup contact) on intentions to help Syrian refugee children [78]. Yet, to our knowledge, there has been no research on how opportunities for direct intergroup contact impact (host society) children's attitudes towards refugee children in Germany.

## The current study

We investigated how 5- and 6-year-olds in Germany perceive refugee children using a combination of three different tasks. We focused on 5- to 6-year-olds due to prior research suggesting high levels of in-group favouritism in this age group [18, 47, 49]. We combined, for the first time, three different tasks to get a comprehensive picture of young children's attitudes and perceptions: (1) a draw-a-typical-child task, including questions about intentions to interact with depicted children (adapted from: [31, 79]), (2) an intergroup attitude task [65, 77], and (3) a newly developed liking task. Similar to previous research [38, 65, 77], we compared children's perceptions of the majority group ("German children") to their perceptions of the minority group ("refugee children"). To investigate potential effects of opportunities for intergroup contact [66, 67, 69], we recruited participants from kindergartens with and without refugee children [66, 67, 69]. The study took place from early 2018 to mid-2019 and hence after the arrival of relatively large numbers of refugees in Germany [14].

First, to broadly investigate pre-schoolers perceptions of refugee children, we adapted the Human-Figure-Drawing task [31, 79] into a *draw-a-typical-child* task. We also used the drawings to ask a series of questions about children's intentions to interact with the depicted child [79]. If children showed favouritism towards the majority group, they would be more willing to interact with German children than with refugee children.

Second, we applied the *intergroup attitude task*, in which children indicated how many members of the minority and majority group, possessed different positive and negative traits such as being nice, unfriendly etc. [65, 77]. If children favoured the majority-group, they would attribute more positive and less negative trait ratings to German children than to refugee children. Furthermore, we used this task to investigate group variability (out-group homogeneity effect) as a less researched aspect of intergroup attitudes [80], especially in young children. We proposed that if participants viewed German children as a more heterogeneous group than refugee children (out-group homogeneity effect), [35, 36, 42] their trait ratings of German children would include fewer extreme responses than their ratings of refugee children.

Third, we tested a new measure, the *liking task*. If children favoured the majority group, they would show greater liking of German children as compared to refugee children.

Last, we recruited participants from kindergartens with and without refugee children to investigate potential effects of opportunities for direct intergroup contact [66, 67, 69]. We expected that children without direct contact opportunity in their kindergarten would show greater differences in the three tasks.

## Materials and methods

### Participants

Sixty 5- to 6-year-old children (Mean age = 5.8 years, SD = 0.3 years, 26 female) from Berlin, Germany, took part in this study. 32 (53%) children attended kindergartens with refugee children and 28 (47%) attended kindergartens without refugee children (for further details, see section "Intergroup contact and contact opportunities"). The sample size of 30 children per contact group was chosen based on previous studies [77]. No *a priori* power analyses were conducted.

Berlin is located in North East Germany and is the largest city in Germany with 3.75 million inhabitants [81]. We recruited children from eleven kindergartens in five different districts of Berlin (see S1 Table in S1 File). During recruitment in kindergartens, we asked pre-school teachers to hand out consent forms to parents of children without refugee status and with sufficient German skills to take part in the study. We included children irrespective of migration history (i.e. irrespective of whether they or their parents were born abroad) and did not collect any data on children's migration history. For conciseness, we will use the term "German children" in the following sections to describe our sample. In each kindergarten, we tested all children with parental consent who were present on testing days. Ten additional children were excluded from the sample: four children had difficulties following instructions, two lost motivation, three had parents who had sought refuge, and one child was dropped due to experimenter error. The main data collection took place from February 2018 to May 2019. In December 2017, ten additional children from a twelfth kindergarten took part in pilot test sessions to ensure the materials and procedure were well understood by children (data not included in data analysis).

### Ethics

The ethics committee of the Department for Education Sciences and Psychology at Freie Universität Berlin approved this study (approval no. 171/2018). Parents provided written, informed consent before their child took part in the study and children gave their spoken assent. Participants/parents received no rewards for participation.

### Intergroup contact and contact opportunities

For our study, we recruited kindergartens that were attended by refugee children and those that were not. This allowed us to assess potential effects of opportunities for contact with refugee peers on our participants' attitudes and perceptions. Seven kindergartens were attended by refugee children, in one kindergarten refugee children attended some of the day-care groups (the groups did not mix), and three kindergartens were not attended by refugee children. Overall, 32 of 60 children had opportunities for direct contact with refugee children in their kindergarten and we used this contact measure (refugee children in kindergarten: yes/no) in all analyses. 60% of 5- to 6-year-olds across contact conditions (71% of those without and 53% of those with contact opportunity in the kindergarten) claimed they had never heard the term "Flüchtling" (German for "refugee") when asked at the beginning of the testing session (see section "Design and procedure").

**Table 1. Interrelationships of contact variables.**

| Measure | 1. | 2. | 3. | 4. | 5. |
|---|---|---|---|---|---|
| 1. sampling: kindergarten attended by refugee children | | | | | |
| 2. parent-report: child's contact to refugee children[A] | .40*** | | | | |
| 3. parent-report: child's contact to refugee adults[A] | .20* | .40*** | | | |
| 4. child-report: child's contact to refugees | .40*** | .13 | .05 | | |
| 5. child-report: familiarity with the term "refugee"[B] | .19* | .20* | .20* | .20* | |

*Note*. All numbers are Kendall's tau ($\tau$) correlation coefficients, except for point-biserial Pearson correlations for contact opportunity (yes/no) in kindergarten.

*p < .05,

**p < .01,

***p < .001.

[A] Parents rated children's contact on a 5-level Likert scale ranging from never to very often (see S1 File).

[B] 30% of children reported to know the term "refugee" (parent-report contact "yes": 38%; parent-report contact "no": 23%).

To get insight into children's and families' general knowledge of and contact with refugees, we asked parents to fill in a questionnaire. The questionnaire for parents included questions such as "How often has your child had contact with refugees until now?" or "How often do you talk about topics like 'displacement', 'refugees' and so on with your child?". Parents responded on a 5-point Likert scale, ranging from "never" to "very often" (for details, see S2 Table in S1 File). We also asked children at the end of test session how many refugees they knew and if they could say their first names. During the study, we noticed that some children in kindergartens with contact opportunities did not know that some of their peers were refugees. For example, some kindergarten teachers reported that they avoided speaking about refugees with children (e.g., to avoid stigmatization of refugee children). However, we found significant correlations between contact opportunity in kindergartens and three other contact measures including both children's and parents' reports of contact to refugees (see Table 1). This suggests that our kindergarten contact measure was a good proxy for overall contact with refugees.

## Design and procedure

The study used a 2 (contact opportunity [between-subject]: yes/ no) x 2 (group condition [within-subject]: German children/refugee children) design.

One experimenter (E) administered the tasks to individual children in a quiet room in the kindergarten. Three different experimenters (two female, one male) collected data for this study. As warm-up, E played with the child in their kindergarten group. Before testing, E informed children about the study being videotaped and asked for their assent. Tasks were administered in fixed order: (1) *draw-a-typical-child task*, (2) *intergroup attitude task*, and (3) *liking task*. Participants completed each task for both groups (refugee, German), before moving on to the next task. Order of groups was identical across tasks but counterbalanced across children and experimenters.

**Draw-a-typical-child task.** We adapted Teichman's Human-Figure-Drawing task [31, 79] into a *draw-a-typical-child* task to examine similarities and differences in participants' perceptions of refugee peers. To this end, participants drew two pictures: a "typical" refugee child and a "typical" German child. During piloting, both groups of three children and single

children performed the draw-a-typical-child task with an experimenter. In the single child setting fewer interruptions occurred and children were able to focus more on the task.

For each drawing, children received an A4-paper and ten coloured pens (yellow, orange, pink, red, purple, grey, green, blue, brown, black). Before participants drew the refugee child, E asked whether they knew the word "refugee" (German: "Flüchtling") and what it meant. The German term "Flüchtling" is closer in meaning to "fugitive" or "runaway" than to "refugee" [82] and can have a negative connotation. Alternative terms like "Geflüchtete" [83] or "Person mit Fluchterfahrung" (engl.: person with experience of flight) are more inclusive, but the term "Flüchtling" is still frequently used in every-day German. As we expected children to be more familiar with this every-day term, we used it in our study. Irrespective of children's responses and, thus, familiarity of the term "Flüchtling", E always provided an explanation of the term "refugeee" ("People who had to flee from their home countries because it was no longer safe for them and some of them now live in Germany."). We did not further probe children's understanding of the term to avoid priming children to think about them as an out-group. Children had up to five minutes (and, if needed, two additional minutes), visualized with an hourglass or a paper clock, to draw each picture. We placed completed drawings out of sight until the end of the drawing task.

We also asked a series of questions about the depicted children. First, E ensured that participants correctly identified the child in the drawing as German/refugee child. Next, E asked a series of questions [84, 85]: (1) 'Does the child have a name?', (2) 'Is [name] a boy or a girl?', (3) 'Would you like to invite [name] to your home?' [Would you like that a bit or a lot?/ Do you like that little or not at all?], (4) 'Would you like to play with [name]?' [same follow up as (3)], and (5) 'Would you like to be as [name]?' [same follow up as (3)]. We always asked questions 1–2 in the same order but counterbalanced the order of questions 3–5 across children.

**Intergroup attitude task.** For the *intergroup attitude task*, children indicated how many members of the minority and majority group possessed different positive and negative traits such as being nice, unfriendly etc. [65, 77]. We were interested in participants' positive and negative ratings as measures of group favouritism [43, 55, 77, 86] and in the number of extreme ratings ("all children" or "no children" are [trait]) as a proxy for perceived group variability [87].

Children saw seven positive and seven negative trait adjectives that were taken from the Preschool Racial Attitude Measure–II (PRAM–II) Series A [32, 77]. We translated all adjectives from English to German including back-translations to ensure accuracy (see S3 Table in S1 File). Doyle and Aboud [55] suggested to present the adjectives with behavioural examples, but piloting revealed that children found this confusing and exhausting as it included a lot of verbal information. Instead, we created small drawings (see S1 Fig in S1 File) and decided to only provide behavioural examples (see S3 Table in S1 File) if children failed to understand an adjective. However, no child required further explanation of the adjectives and we thus never used the behavioural examples.

During the task, E drew all adjective cards randomly from a cloth bag. We measured participants' responses using a stick-people scale [42, 43, 51, 77], ranging from "all children", "most children", "half of the children", "a few children", to "no children" (Fig 1). E introduced the scale by placing the stick-people cards in the right order, then shuffled them and asked participants to arrange them correctly again. Next, E asked a series of practice questions such as "How many children like to eat chocolate?", pointing to each of the possible responses ("All children?", "Most children?" etc.).

For the main task, E showed participants two photo collages [77]. One collage included pictures of children from different ethnic minority groups in Germany (i.e., Central and Southern Asian, Northern and Central African, Eastern European) who were introduced as refugee

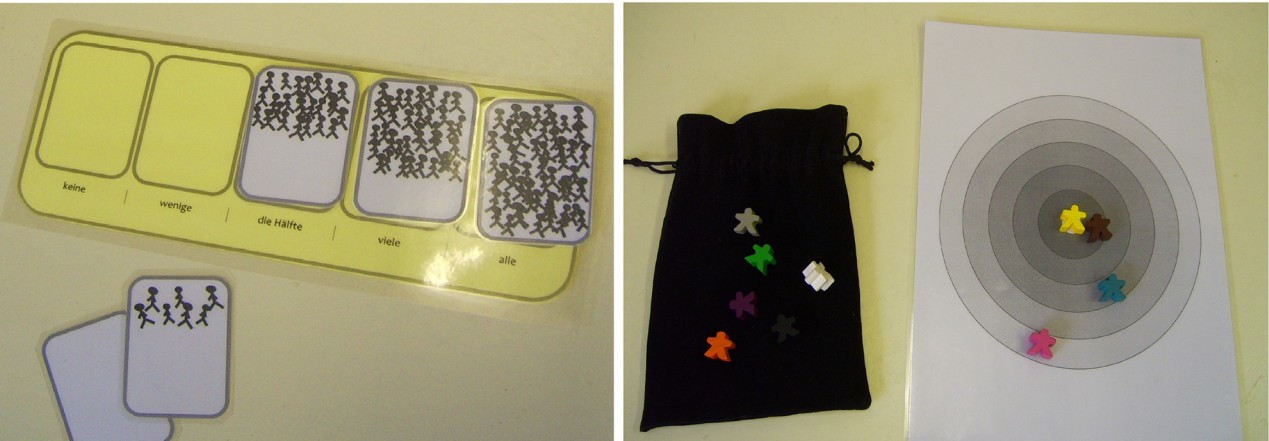

**Fig 1. Screenshots of the experimental interview set-up for the intergroup attitude task (left) and the liking task (right).**

children (German: "Flüchtlingskinder"). The other collage included pictures of children from mostly European ethnic majority groups who were introduced as German children (German: "deutsche Kinder"). To test participants' associations, E randomly retrieved an adjective card from the bag, placed it on the collage and stated, "The word is [adjective]. How many refugee/ German children are [adjective]?" Children answered using the stick-people scale. E repeated this for all adjectives. To ensure that participants remained focused and engaged, they played a simple game between conditions.

**Liking task.** The development of the liking task was inspired by the OSIO (Overlap of Self, In-group, and Outgroup) measure [88] and socio-metric methods used to investigate pre-schoolers peer preferences [89]. In our study, children used a game board with five concentric circles (Fig 1) to indicate their liking of refugee peers and German peers (represented by small wooden figures) through spatial proximity on the board. Piloting revealed children had difficulties understanding the relation between spatial proximity of the figures on the board and liking of the persons they represented. We therefore included a short comprehension task.

To introduce the task, E placed a figure of themselves in the centre and three figures of people they liked a lot, they somewhat liked, and they did not like at all increasingly further away from the centre (innermost circle: like a lot, middle circle: somewhat like, outermost circle: do not like at all). E then removed all figures from the board, gave children their own figure and three additional ones (someone they did not like, someone they liked a little bit, someone they liked a lot). Children continued to the main task when they successfully placed the figures in the correct relative order on the circles.

For the main task, E removed all figures—apart from the participant's own—and gave participants two new figures to place on the board: the participant's best friend and someone they rarely played with. Next, E showed the collage of the refugee/German children and handed the participant a figure representing the respective target child. Participants were instructed to place the figure on the board so that one can see how much they like the refugee/German child. E then removed this figure and repeated the procedure for the other target child.

**End of the testing session.** At the end of the session, E asked participants if they knew any refugees and their first names. We included these questions as some children claimed during piloting that they knew a refugee child from TV. Furthermore, we asked children for permission to keep their drawings (two children denied permission and we photographed their

drawings). We also conducted group-debriefing sessions with 2–6 participants each, in which children could ask questions and talk about the study.

## Data coding

E live-coded children's answers. A coder (coder 1; none of the authors) later checked the coding for accuracy using the video recordings (except for two children without consent for videotaping).

**Draw-a-typical-child task.** We coded children's drawings with a mixed deductive and inductive approach [90], using Teichman's work [31] as a starting point. Specifically, in a first step the coder scored the number of colours in the drawing, the number of items, the size of the figure, and the complexity of the figure. The coder also rated whether the figure's face and the general environment appeared positive, neutral or negative (for details see S4 Table in S1 File). Coder 1 scored the drawings blind to group condition (German or refugee child) in random order. In step 2, the coder scored the overall similarity of the two drawings for each child on a 4-point Likert scale (1—very different, 2—rather different, 3—rather alike, 4—very alike). For the global similarity rating, we rearranged each participant's two drawings side-by-side (counterbalanced for group). A second coder scored 25% of the drawings for reliability purposes. Agreement between coders was excellent [91] for size (unweighted-κ = 1.00; we counted rating differences below 0.5 SD as agreement because slight variations already had a strong impact on the reliability coefficient), number of items in the drawing (equal-weighted-κ = 0.85), and the affect in the face of the depicted figure (equal-weighted-κ = 0.86). Agreement was good for number of colours in the drawing (equal-weighted-κ = 0.76), complexity of the figure (equal-weighted-κ = 0.77), global similarity (equal-weighted κ = 0.76), and environment ratings (equal-weighted-κ = 0.60).

We calculated a composite score for participants' intentions to interact with the respective target child in the drawings by adding up the answers to the three questions (e.g. as 'Would you like to play with [child]?'), resulting in a sum score ranging from four to 12.

**Intergroup attitude task.** We analysed both *attitude scores* and the number of extreme ratings as an indicator for *group variability*. First, we summed participants' ratings (5-point Likert scale) of the seven positive traits (range: 7 to 35) and of the seven negative traits (range: -7 to -35) for refugee and German children, respectively [77]. For reliabilities of pre-scores, see S1 File. We then summed the positive and the negative attitude scores to arrive at an overall attitude score for each group (range = -28 to +28). Second, we summed the number of participants' extreme responses ("no children" and "all children") when rating German and refugee children, respectively. Higher numbers of extreme ratings indicate less variable (more homogenous) attitudes towards the group [42, 43].

**Liking task.** We recorded each figure's position on the board as a 5-point Likert scale, with the innermost circle coded as 5 and the outermost circle as 1.

## Data analysis

For each of our measures, we performed both a frequentist Anova, using the afex package [92] and a Bayesian ANOVA with one within-subjects factor (group) and two between-subject factors (contact opportunity and gender), using the BayesFactor package [93] in R (Version 3.6.3) [94]. We used frequentist and Bayesian approaches side-by-side in our analyses to get a more complete picture of the data and to offer readers the choice to interpret results in both statistical frameworks. Our analysis of global similarity of the drawings (Draw-a-typical-child task) only included group (within-subject factor) and gender (between-subjects factor). We considered Bayes factors smaller than 0.3 as evidence for the null hypothesis and Bayes factors larger

than 3.0 as evidence for the alternative hypothesis [95, 96]. For details on handling of missing values, see S1 File.

## Results

### Draw-a-typical-child task

First, we analysed whether participants' drawings of refugee and German children differed on any of the six coding categories (e.g., *number of items in the drawing, affect of the depicted figure's face;* for descriptive statistics see S6 Table in S1 File). For all coding categories, we neither found significant main effects of group or contact, nor significant interaction effects (see S7 Table in S1 File for detailed outputs). However, our analyses revealed gender effects for some of the coding categories. Girls used significantly more colours in their drawings then boys, $F(1, 57) = 28.44$, $p = < 0.001$, $\eta^2_g = 0.286$, $BF_{10} = 7005.06$. They also drew the affect in the figure's face significantly more positive than boys, $F(1, 57) = 8.75$, $p = 0.005$, $\eta^2_g = 0.077$, but the Bayes Factor $BF_{10} = 1.81$, indicated that the data were insensitive (offering neither support for $H_0$ nor for $H_1$). There was a significant main effect of gender for the depicted environment, $F(1, 57) = 9.63$, $p = 0.003$, $\eta^2_g = 0.117$, $BF_{10} = 10.95$. We also found a significant interaction of gender*group, $F(1, 57) = 4.39$, $p = 0.041$, $\eta^2_g = 0.017$, but $BF_{10} = 1.75$ indicated the data were insensitive. Next, we compared the global similarity ratings for participants' drawings of German and refugee children (see Table 2 for mean values and standard deviations). Consistent with the previous results, there was no significant main effect of contact opportunity, $F(1, 57) = 0.11$, $p = 0.742$, $\eta^2_g = 0.002$, with $BF_{10} = 0.27$ indicating substantial support for $H_0$ (global similarity is the same in both contact groups). There was also no significant main effect of gender $F(1, 57) = 0.24$, $p = 0.629$, $\eta^2_g = 0.004$, with $BF_{10} = 0.28$ indicating evidence for the $H_0$. We also analysed participants' intentions to interact with the depicted children in the drawings and found no significant main effect for group, $F(1, 57) = 0.67$, $p = 0.418$, $\eta^2_g = 0.003$, with $BF_{10} = 0.28$ indicating evidence for the $H_0$ (equal intentions to interact with German and refugee children). We also found no significant main effect of contact opportunity, $F(1, 57) = 3.12$, $p = 0.083$, $\eta^2_g = 0.039$, $BF_{10} = 1.22$, or gender, $F(1, 57) = 0.00$, $p = 0.978$, $\eta^2_g = <0.001$, $BF_{10} = 0.35$ –both Bayes factors indicating data insensitivity. There was also no significant interaction both between contact opportunity*group, $F(1, 57) = 2.55$, $p = 0.116$, $\eta^2_g = 0.011$, $BF_{10} = 0.76$, and between gender*group, $F(1,57) = 0.01$, $p = 0.924$, $\eta^2_g = < 0.001$, with $BF_{10} = 0.28$ indicating evidence for the $H_0$.

**Table 2. Descriptive statistics (Mean [M], Standard Deviation [SD]) for the draw-a-typical-child task ($n_{contact\ opportunity}$ = 32; $n_{no\ contact\ opportunity}$ = 28).**

| | | Contact opportunity | | No contact opportunity | |
|---|---|---|---|---|---|
| | | **M** | **(SD)** | **M** | **(SD)** |
| Global similarity (4-point Likert scale)[A] | | 2.91 | (0.95) | 2.96 | (1.22) |
| Intentions to interact with depicted children child in picture (3-point Likert scale)[B] | German child | 3.20 | (0.80) | 2.61 | (0.98) |
| | Refugee child | 2.90 | (1.12) | 2.70 | (0.81) |
| Drawing includes target figure | German child | 32 of 32 | | 27 of 28 | |
| | Refugee child | 31 of 32 | | 27 of 28 | |

Note.

[A]With higher ratings indicating greater similarity (range from 1-very much different to 4-very much alike).

[B]With higher ratings indication greater positivity.

**Table 3. Descriptive statistics (Mean [M], Standard Deviation [SD]) for the intergroup attitude task ($n_{\text{contact opportunity}}$ = 32; $n_{\text{nocontact}}$ = 28).**

|  |  | All children | | Contact opportunity | | No contact opportunity | |
|---|---|---|---|---|---|---|---|
|  |  | **M** | **(SD)** | **M** | **(SD)** | **M** | **(SD)** |
| Group favouritism | German children | 8.78 | (8.95) | 8.03 | (9.00) | 9.64 | (8.97) |
|  | Refugee children | 8.61 | (9.50) | 6.91 | (9.04) | 10.57 | (9.80) |
| Group variability (homogeneity) | German children | 5.32 | (3.61) | 4.94 | (3.47) | 5.75 | (3.79) |
|  | Refugee children | 6.18 | (3.63) | 5.84 | (3.03) | 6.57 | (4.24) |

Note. Group favouritism scores range from -21 to +21. Group variability: smaller no. of extreme ratings (all children/ no children) indicates greater group variability (less group homogeneity).

### Intergroup attitude task

Table 3 summarizes means and standard deviations of values in the intergroup attitude task. Our analyses of children's group favouritism found no significant main effect of group $F(1, 57)$ < 0.001, p = 0.960, $\eta^2_g$ = <0.001, with $BF_{10}$ = 0.20 indicating evidence for $H_0$ (similar attitudes towards both groups). We found no significant main effect of contact, $F(1, 57)$ = 2.86, p = 0.096, $\eta^2_g$ = 0.037, $BF_{10}$ = 0.61, or gender, $F(1, 57)$ = 2.79, p = 0.100, $\eta^2_g$ = 0.036, $BF_{10}$ = 0.59. Results also revealed no significant interaction between group*contact, $F(1, 57)$ = 1.25, p = 0.268, $\eta^2_g$ = 0.005, $BF_{10}$ = 0.38, or between group*gender, $F(1, 57)$ = 0.75, p = 0.389, $\eta^2_g$ = 0.003, $BF_{10}$ = 0.30; Bayes factors indicate data insensitivity.

In addition, we analysed children's extreme ratings (all children, no children) in the intergroup attitude task as an indicator of group variability (homogeneity). There was a significant main effect of group, $F(1, 57)$ = 5.51, p = 0.022, $\eta^2_g$ = 0.013, with $BF_{10}$ = 2.80 suggesting moderate evidence that children viewed refugee children as a more homogenous group than German children. There was no significant main effect of contact, $F(1, 57)$ = 1.10, p = 0.298, $\eta^2_g$ = 0.016, $BF_{10}$ = 0.51, or gender, $F(1, 57)$ = 0.54, p = 0.464, $\eta^2_g$ = 0.008, $BF_{10}$ = 0.42. We also found no interaction between group*contact, $F(1, 57)$ = 0.05, p = 0.825, $\eta^2_g$ = <0.001, $BF_{10}$ = 0.25, or between group*gender, $F(1, 57)$ = 0.15, p = 0.702, $\eta^2_g$ = <0.001, $BF_{10}$ = 0.28; Bayes factors for both results indicate evidence for $H_0$.

### Liking task

Finally, we compared children's liking scores and found a significant main effect of group, $F(1, 54)$ = 6.73, p = 0.012, $\eta^2_g$ = 0.040, with $BF_{10}$ = 5.20, suggesting evidence for the $H_1$ (participants liked German children more than refugee children). There was no significant main effect of contact $F(1, 54)$ = 0.06, p = 0.805, $\eta^2_g$ = <0.001, $BF_{10}$ = 0.35, or gender, $F(1, 54)$ = 3.25, p = .0.077, $\eta^2_g$ = 0.039, $BF_{10}$ = 1.28, both results indicating data insensitivity. Our analyses did not reveal a significant interaction between group*contact, $F(1, 54)$ = 0.02, p = 0.878, $\eta^2_g$ = <0.001, with $BF_{10}$ = 0.23 indicating evidence for $H_0$ (no interaction of group and contact). Moreover, there was no significant interaction between group*gender, $F(1, 54)$ = 0.09, p = 0.771, $\eta^2_g$ = <0.001, $BF_{10}$ = 0.27 (evidence for $H_0$). Table 4 summarizes means and standard deviations for the *liking task*.

### Discussion

An increasing number of researchers have highlighted the need to extend studies of intergroup attitudes beyond minimal group paradigms to real-world settings that include macro-contextual variables such as historical and geographical factors [18–20]. Our research addresses this

**Table 4. Descriptive statistics (Mean [M], Standard Deviation [SD]) for the liking task ($n_{contact\ opportunity}$ = 31; $n_{no\ contact\ opportunity}$ = 26)[A].**

| | | All children | | Contact opportunity | | No contact opportunity | |
|---|---|---|---|---|---|---|---|
| | | *M* | *(SD)* | *M* | *(SD)* | *M* | *(SD)* |
| **Group liking** | **German children** | 2.63 | (1.44) | 2.74 | (1.53) | 2.50 | (1.36) |
| | **Refugee children** | 2.04 | (1.59) | 2.16 | (1.66) | 1.88 | (1.53) |

*Note.* Liking ranges from 1 (not like at all) to 5 (like a lot).

[A]Due to experimenter error, data of three participants were not included into the analyses of this task (n = 57). Greater scores indicate more liking.

gap in the literature by investigating 5- to 6-year-olds' perceptions of refugee children in the historical context of the so-called refugee crisis in Germany. Specifically, children in our study completed a set of three tasks consisting of a *draw-a-typical-child task* [31], an *intergroup attitude task* [77], and a newly developed *liking task*. In all tasks, we compared their perceptions of German children (majority group) with their perceptions of refugee children (minority group). Our results revealed that participants drew similar pictures of German and refugee children and had similar intentions to interact with the depicted children. Moreover, we found evidence that children did not favour either group in the intergroup attitude task (explicit measure). However, participants favoured German children over refugee children in the liking task and rated refugee children as a more homogenous group than German children (based on extreme ratings "all"/"none" in the attitude task). Opportunities for direct contact (in the kindergarten) had no discernible effect on any of our measures. Developmental research has shown a peak in in-group favouritism at pre-school age [31, 46–48, 51, 97, 98] and a gradual suppression of explicit (but consistent levels of implicit) bias towards middle childhood [86]. Contrary to these previous findings, 5- to 6-year-old children in our study showed no favouritism of German children as compared to refugee children in the intergroup attitude task [46]. However, they favoured German children in the liking task. It is possible that children in our study only had a vague perception of refugee children as "others" whom they liked less. Our observation that 60% of 5- to 6-year-olds across contact conditions (71% of those without and 53% of those with contact opportunity in the kindergarten) initially claimed they had not heard the term "Flüchtling" (German for "refugee") corroborates this interpretation. It should be noted though that all children were provided with an explanation of the term "refugee" before completing the three tasks. Moreover, a meta-study [99] has shown that mere exposure effects occur even in the absence of stimulus recognition and that stimulus recognition might even impede positive exposure effects. Thus, knowing the term "refugee" may not be necessary for children to show in-group favouritism. Nevertheless, future research on attitudes towards real-world groups like refugees could include qualitative methods, such as interviews or group discussions, to explore children's views on the respective group in more detail.

Developmental research that has found differences in children's views of minority and majority group members has usually been conducted in the context of long-lasting and deeply rooted conflicts such as the Arab-Jewish conflict in Israel [30, 31, 53]. In contrast, the arrival of large numbers of refugees is still a recent phenomenon in Germany and, despite a prominent public discourse about refugees at the time of the study, it is possible that "refugees" are not a salient enough minority group category to influence young children's explicit attitudes.

Analyses of extreme ratings in the intergroup attitude task (e.g., "all" or "no" children) showed that children in our study used extreme ratings more frequently in relation to refugee

peers as compared to German peers. This suggests that they viewed refugee children as a more homogenous/less variable group than German children (group homogeneity effect towards refugee children) [37, 66, 87]. Our findings are in line with previous research from Canada on "pro-White/anti-Black biases" that showed an increase in perceived out-group homogeneity from 3 to 6 years of age [44]. There is evidence that perceived out-group homogeneity (variability) is related to intergroup biases and discrimination and even dehumanization [100]. Studies with adults have found that interventions aimed at increasing perceptions of minority group variability reduced intergroup bias [36, 101]. Future developmental research could investigate whether group variability interventions would also reduce prejudice in children, for instance, by conducting a longitudinal intervention study.

Previous work has usually relied on a single task to measure children's intergroup biases [74, 102]. We combined an established intergroup attitude task [77] and a variant of a draw-a-typical-human task [31] with our newly developed liking task. The liking task has the advantage that it can be implemented quickly and easily and is less verbally taxing than, for instance, the intergroup attitude task. This new measure could be useful for longitudinal investigations or could be utilized in combination with qualitative methods to further explore young children's intergroup attitudes.

In our study, we investigated the role of direct intergroup contact opportunities by sampling from kindergartens with and without refugee children. We found no effects of direct contact opportunities on participants' drawings, homogeneity ratings, and group favouritism. This is in contrast to prior research that suggested positive effects of contact on intergroup attitudes [67, 101, 103], such as direct contact interventions [76] and mere exposure [70, 99], for instance, in school contexts [75]. Recently, research with adults has started to focus on negative or neutral effects of intergroup contact [104, 105]. Future developmental research could investigate the quality of children's intergroup contact experiences and preferences in a qualitative manner, which could also yield new insights for designing new contact interventions [105, 106].

It is possible that the study location (Berlin), a cosmopolitan and heterogeneous city where for instance 18% of the population do not have German citizenship (see S1 Table in S1 File), could explain the lack of direct contact effects in our study as previous research has found that contact has stronger effects in less diverse settings [107]. Moreover, during the study, educational staff in some kindergartens attended by refugee children informed us that the refugee status of children in their care was not common knowledge. Some institutions even avoided talking about topics such as displacement because they were worried about stigmatizing refugee children and of re-traumatizing them. In addition, we used opportunities for direct contact as a binary measure (yes/no) and did not assess the quantity nor the quality of intergroup contact in the kindergarten [108–110]. Future developmental research on group attitudes in real-world settings could consider these contextual factors, for instance, by investigating both quantity and quality of intergroup contact and/or by sampling from more and less diverse regions within a country. Furthermore, future studies could investigate micro-variations across institutions (schools/kindergartens) in how they address questions of diversity and refugees and whether they actively address the refugee status of children in their care.

Most research to date has focused on majority members' attitudes of minority groups and on positive effects of intergroup contact, yet there are increasing calls to include different perspectives and voices in the study of intergroup attitudes [19]. For example, future research could investigate contact experiences of both refugees and majority group members in different age groups. Importantly, such research would need to take the diversity of refugee communities and, thus, potential differences in their perspectives and attitudes into account.

Studies on the development of intergroup attitudes in real-world settings can play an important role in informing pedagogic approaches and policies. Research on children's socio-cognitive development has overwhelmingly shown that social categorization is a developmental process that allows children to make sense of their social world [30, 37, 111] and forms the basis for perceived group variability and group perceptions [28]. In our study, we found an out-group homogeneity bias towards refugee peers in young children. This may inform pedagogical approaches that encourage children to reflect upon their own and others' group membership(s) and to think about the variability within social groups (e.g. gender, ethnicity, social origins).

Forecasts suggest that more people than ever are going to migrate in the future due to the compounded effects of the climate emergency, pandemics (like the COVID-19 pandemic), wars, and famines [112, 113]. At the time of writing this article in 2022, the Russian invasion of Ukraine has resulted in millions of displaced Ukrainians across Europe. Food shortages due to blockage, delays of grain exports and the ongoing energy crisis are projected to cause social and political upheaval in countries across the world [114, 115]. Understanding how intergroup attitudes are formed during childhood and impact social behaviours and, ultimately, public policy appears more pressing than ever.

## Supporting information

**S1 File. Supporting information for "pre-schoolers' perceptions of refugee children".** (DOCX)

## Acknowledgments

We thank Daniil Serko and Kathrin Kern for their help with data collection and data entry, and Elena Schmidt Velasco for her help with data coding.

We further thank Flavio Morelli (fu:stat; Freie Universität Berlin) for statistical advise and Inka Bormann for general feedback on the project.

We are grateful for the valuable contribution of all participating children, parents, and kindergartens.

## Author Contributions

**Conceptualization:** Iris Würbel, Patricia Kanngiesser.

**Formal analysis:** Iris Würbel.

**Investigation:** Iris Würbel.

**Methodology:** Iris Würbel, Patricia Kanngiesser.

**Writing – original draft:** Iris Würbel.

**Writing – review & editing:** Patricia Kanngiesser.

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
