## [Decision Letter · Decision Letter 0]

29 Aug 2022

PONE-D-22-20046Pre-schoolers’ perceptions of refugee childrenPLOS ONE

Dear Dr. Würbel,

Thank you for submitting your manuscript to PLOS ONE. After careful consideration, we feel that it has merit but does not fully meet PLOS ONE’s publication criteria as it currently stands. Therefore, we invite you to submit a revised version of the manuscript that addresses the points raised during the review process.

We look forward to receiving your revised manuscript.

Kind regards,

Sawsan Abuhammad

Academic Editor

PLOS ONE

Journal Requirements:

“Patricia Kanngiesser was supported by a Freigeist Fellowship from Volkswagen Foundation.”

“Patricia Kanngiesser was supported by a Freigeist Fellowship from Volkswagen Foundation. The funders had no role in study design, data collection and analysis, decision to publish, or preparation of the manuscript.”

Additional Editor Comments:

The manuscript PONE-D-22-20046 describes an experimental study examining pre-schoolers’ perceptions of refugee children among a German sample. The major strength of the paper is that it draws upon established intergroup theories for an important, current topical social behaviour. The main issues I have with the paper lie in (1) the need to cover some of the basic, classic theorizing of intergroup behaviours even if briefly and (2) the need to align the final practical applications more closely with the study’s findings. These points, along with other issues, are detailed below.

Introduction

1. The title is a little brief and could be expanded to reflect more of the study detail.

2. Some of the wording is hard to understand such as “some scholars ask for a critically use of the term..”. (p. 1).

3. Could refugee crisis be either placed in quotes or be preceded by ‘so called’ without both measures unless it is essential to do so? (p. 1 and throughout manuscript)

4. Does the term “Hooligans” need more explanation to ensure understanding from a wide audience? (p. 1)

5. Would the theoretical background benefit from citing a social identity/self-categorisation theory approach?

6. Again, to ensure wide audience understanding, can the minimal group paradigms reference please be explained briefly with citation? (p. 4)

7. There is a lot of detail about the measures in the Introduction section that may be useful in this section but could be consolidated more in the Method section?

8. My understanding is references in brackets should be in alphabetical order (throughout manuscript) and that “and colleagues” requires a citation in brackets to accompany it (e.g., p. 5).

Method and Results

9. Table 1: there is no need to repeat the correlation information twice (above and below the table diagonal) (p. 9)

10. Could some examples be provided for ‘behavioural examples’ (p. 12)?

11. Could more information be provided about the piloting that occurred? (p. 12)

12. “data was” should be “data were” (p. 16 – twice)

Discussion

13. There is some confusing wording with “not knowing the term “refugee”” (p. 20)

14. The applied implications referring to encouraging the appreciation of differences etc should be more explicitly linked with the study’s findings. Does it link to the finding about liking German students more than refugee children? (p. 23)

15. Given the study’s significant findings, it may be beneficial, as part of the applied implications, to encourage thinking about homogeneity/heterogeneity (and that all labelled groups comprise diversity within them) and how that might be translated into intervention strategies? (p. 23)

16. If the Anti-Bias approach is considered feasible and aligned with the study’s findings, can a little more information please be provided about it? (p. 23)

Recommendation:

The stated points are easily addressed and the manuscript is a very worthwhile contribution to the extant literature. The paper contributes usefully to our understanding of this important and topical area.

Reviewers' comments:

Reviewer's Responses to Questions

**Comments to the Author**

1. Is the manuscript technically sound, and do the data support the conclusions?

Reviewer #1: Yes

2. Has the statistical analysis been performed appropriately and rigorously? 

Reviewer #1: Yes

3. Have the authors made all data underlying the findings in their manuscript fully available?

Reviewer #1: No

4. Is the manuscript presented in an intelligible fashion and written in standard English?

Reviewer #1: Yes

5. Review Comments to the Author

Reviewer #1: The manuscript PONE-D-22-20046 describes an experimental study examining pre-schoolers’ perceptions of refugee children among a German sample. The major strength of the paper is that it draws upon established intergroup theories for an important, current topical social behaviour. The main issues I have with the paper lie in (1) the need to cover some of the basic, classic theorizing of intergroup behaviours even if briefly and (2) the need to align the final practical applications more closely with the study’s findings. These points, along with other issues, are detailed below.

Introduction

1. The title is a little brief and could be expanded to reflect more of the study detail.

2. Some of the wording is hard to understand such as “some scholars ask for a critically use of the term..”. (p. 1).

3. Could refugee crisis be either placed in quotes or be preceded by ‘so called’ without both measures unless it is essential to do so? (p. 1 and throughout manuscript)

4. Does the term “Hooligans” need more explanation to ensure understanding from a wide audience? (p. 1)

5. Would the theoretical background benefit from citing a social identity/self-categorisation theory approach?

6. Again, to ensure wide audience understanding, can the minimal group paradigms reference please be explained briefly with citation? (p. 4)

7. There is a lot of detail about the measures in the Introduction section that may be useful in this section but could be consolidated more in the Method section?

8. My understanding is references in brackets should be in alphabetical order (throughout manuscript) and that “and colleagues” requires a citation in brackets to accompany it (e.g., p. 5).

Method and Results

9. Table 1: there is no need to repeat the correlation information twice (above and below the table diagonal) (p. 9)

10. Could some examples be provided for ‘behavioural examples’ (p. 12)?

11. Could more information be provided about the piloting that occurred? (p. 12)

12. “data was” should be “data were” (p. 16 – twice)

Discussion

13. There is some confusing wording with “not knowing the term “refugee”” (p. 20)

14. The applied implications referring to encouraging the appreciation of differences etc should be more explicitly linked with the study’s findings. Does it link to the finding about liking German students more than refugee children? (p. 23)

15. Given the study’s significant findings, it may be beneficial, as part of the applied implications, to encourage thinking about homogeneity/heterogeneity (and that all labelled groups comprise diversity within them) and how that might be translated into intervention strategies? (p. 23)

16. If the Anti-Bias approach is considered feasible and aligned with the study’s findings, can a little more information please be provided about it? (p. 23)

Recommendation:

The stated points are easily addressed and the manuscript is a very worthwhile contribution to the extant literature. The paper contributes usefully to our understanding of this important and topical area.

6. PLOS authors have the option to publish the peer review history of their article (what does this mean?). If published, this will include your full peer review and any attached files.

Reviewer #1: No

---

## [Author Response · Author response to Decision Letter 0]

28 Dec 2022

PONE-D-22-20046

Pre-schoolers’ perceptions of refugee children

We would like to thank the reviewer and the editor for their valuable and encouraging feedback and appreciate their time and effort in helping us improve our manuscript. 

Below we respond to each of the comments. For convenience, the editor’s and reviewer’s comments are included (shaded in grey) together with our point-by-point responses. 

Kind regards, the authors

Editor’s comments:

Please find the marked-up copy of the manuscript including changes and the unmarked version of the revised paper in the PLOS ONE Editorial Manager Submission System.

>>>>>> We made changes to our financial disclosure and thus updated our cover letter.

>>>>>> We have checked our figure file following your recommendation with PACE. The figure follows the PACE guidelines now. We have updated the file upload in the Submission System respectively. 

If applicable, we recommend that you deposit your laboratory protocols in protocols.io to enhance the reproducibility of your results. Protocols.io assigns your protocol its own identifier (DOI) so that it can be cited independently in the future. For instructions see: https://journals.plos.org/plosone/s/submission-guidelines#loc-laboratory-protocols. 

>>>>>> We appreciate the opportunity to share laboratory protocols. All materials of our study are however included in the Supplement Information (S1_File).

Additionally, PLOS ONE offers an option for publishing peer-reviewed Lab Protocol articles, which describe protocols hosted on protocols.io. Read more information on sharing protocols at https://plos.org/protocols?utm_medium=editorial-email&utm_source=authorletters&utm_campaign=protocols.

>>>>>> Thanks for sharing this valuable information that we will mind for future publications. However, we figure that the frame of Lab Protocol articles does not apply to our current study.

Journal Requirements:

>>>>>> We adopted our manuscript to PLOS ONE´s style requirements for the main body and formatting/sample/title/authors/affiliations.

>>>>>> Thank you for spotting this. We updated the ‘Funding Information’ statement in the Cover Letter as follows (see also comment #3):

“I have read the journal's policy and the authors of this manuscript have the following competing interests: Patricia Kanngiesser was supported by a Freigeist Fellowship from Volkswagen Foundation (grant no.: 89611). The funders had no role in study design, data collection and analysis, decision to publish, or preparation of the manuscript.”

“Patricia Kanngiesser was supported by a Freigeist Fellowship from Volkswagen Foundation.”

“Patricia Kanngiesser was supported by a Freigeist Fellowship from Volkswagen Foundation. The funders had no role in study design, data collection and analysis, decision to publish, or preparation of the manuscript.”

>>>>>> Thank you for your helpful suggestions concerning the funding statement and for updating the online submission form respectively. We removed any funding information from the Acknowledgements Section. Moreover, we would like to suggest this version for the Funding Statement section (also included in the Cover letter, see also comment #2):

“Patricia Kanngiesser was supported by a Freigeist Fellowship from Volkswagen Foundation (grant no.: 89611). The funders had no role in study design, data collection and analysis, decision to publish, or preparation of the manuscript.”

We also included this statement into our cover letter.

>>>>>> Thank you for raising this important issue. We re-checked the approval of Freie Universität’s ethics committee and parents’ consent. As a result, there are both ethical (children as a most vulnerable group whose private data need particular care) and legal restrictions (parents’ did not consent for openly sharing anonymised data of them and of their children) preventing us from sharing data, including the minimal data underlying the results. However, de-identified data are available upon request from co-author Iris Würbel (iris.wuerbel@fu-berlin.de). We now write in the Data availability statement:

“Data cannot be shared publicly because no parental consent was elicited for public sharing of data. This was done due to the potentially sensitive topic of asking children about refugee peers. De-identified data are available upon request from researchers at academic institutions who sign a confidentiality agreement.”

>>>>>> Thanks for mentioning this issue. We checked our reference list and it is complete and correct. We updated references and in-text citations to the reference style outlined by the International Committee of Medical Journal Editors (ICMJE), also referred to as the “Vancouver” style. We did not include retracted papers into our manuscript.

Additional Editor Comments: / 

Reviewers' comments:

The manuscript PONE-D-22-20046 describes an experimental study examining pre-schoolers’ perceptions of refugee children among a German sample. The major strength of the paper is that it draws upon established intergroup theories for an important, current topical social behaviour. The main issues I have with the paper lie in (1) the need to cover some of the basic, classic theorizing of intergroup behaviours even if briefly and (2) the need to align the final practical applications more closely with the study’s findings. These points, along with other issues, are detailed below.

>>>>>> Thank you for your encouraging feedback regarding our manuscript. We describe in detail below how we have responded to each of your suggestions. 

Introduction

1. The title is a little brief and could be expanded to reflect more of the study detail.

>>>>>> Thank you for your suggestion. We have altered both full and short title to better reflect the study location and methods/measures.

We have changed the FULL TITLE to: “Pre-schoolers’ images, intergroup attitudes, and liking of refugee peers in Germany”.

The SHORT TITLE has been updated to: “Pre-schoolers’ perceptions of refugee peers in Germany”.

2. Some of the wording is hard to understand such as “some scholars ask for a critically use of the term.”. (p. 1).

>>>>>> Thank you for pointing out that some of our wording was unclear. We now write (p. 1):

“(Note that we approach the notion of a “refugee crisis” critically as this term often carries a negative connotation [3].)”

3. Could refugee crisis be either placed in quotes or be preceded by ‘so called’ without both measures unless it is essential to do so? (p. 1 and throughout manuscript)

>>>>>> Thank you for spotting this, we have now used the term “so called refugee crisis” throughout the manuscript (pp. 1, 2, 4, 18).

4. Does the term “Hooligans” need more explanation to ensure understanding from a wide audience? (p. 1)

>>>>>> Thanks for pointing this out. We now clarify the term (p. 1):

“(in public discourses, hooliganism refers to verbal and physical violence of football fans [13]”. 

We thereby refer to:

13. Dunning E. Towards a Sociological Understanding of Football Hooliganism as a World Phenomenon. European Journal on Criminal Policy and Research. 2000;8(2):141-62. doi: 10.1023/a:1008773923878.

5. Would the theoretical background benefit from citing a social identity/self-categorisation theory approach?

>>>>>> Thank you for your suggestion to broaden the theoretical background of our paper. Following your recommendations, we now start the sub section “Development of social categorization and intergroup attitudes” with a short introduction of Drew Nesdale’s Social Identity Development Theory. We write (p. 2): 

“Social Identity Development Theory (SIDT) describes how children become increasingly Social Identity Development Theory describes how children become increasingly involved in and aware of social groups [28]: Children between the ages of two to six years increasingly apply social categories to themselves and others, prefer their in-group, and perceive out-groups as homogenous.”

6. Again, to ensure wide audience understanding, can the minimal group paradigms reference please be explained briefly with citation? (p. 4)

>>>>>> Thanks for your suggestion to briefly explain the minimal group paradigm. We now write (p. 3):

“Yet, first observed by Henri Tajfel [40], membership in a “minimal” group based on arbitrary categories, such as blue or red, is sufficient to induce preferences for the in-group [18, 41].”

We refer to the following references:

18. Dunham Y, Baron AS, Carey S. Consequences of “Minimal” Group Affiliations in Children. Child Development. 2011;82(3):793-811. doi: 10.1111/j.1467-8624.2011.01577.x.

40. Tajfel H, Turner JC. The social identity theory of intergroup behavior. In: Jost JT, Sidanius J, editors. Political psychology: Key readings London: Psychology Press; 2004 p. 276-93.

41. Locksley A, Hepburn C, Ortiz V. Social Stereotypes and Judgments of Individuals. Journal of Experimental Social Psychology. 1982;18:23-42. doi: 10.1016/0022-1031(82)90079-8.

7. There is a lot of detail about the measures in the Introduction section that may be useful in this section but could be consolidated more in the Method section?

>>>>>> Thank you for suggesting a consolidation of the measures in the Method section. We have shortened the description of measures in the introduction as much as possible while ensuring that readers are provided with sufficient methodological details to understand our hypotheses (p. 5f.). 

In addition, we have moved details from the introduction on the methods section. Specifically, we now write in the methods section: 

For the “Draw-a-typical task”, we now write (p. 10):

“We adapted Yona Teichman´s Human-Figure-Drawing task [31, 79] into a draw-a-typical-child task to examine similarities and differences in participants’ perceptions of refugee peers. To this end, participants drew two pictures: a “typical” refugee child and a “typical” German child.”

For the „Intergroup attitude task”, we now write (p. 11):

“For the intergroup attitude task, children indicated how many members of the minority and majority group possessed different positive and negative traits such as being nice, unfriendly etc. [65, 77]. We were interested in participants’ positive and negative ratings as measures of group favouritism [43, 55, 77, 86] and in the number of extreme ratings (“all children” or “no children” are [trait]) as a proxy for perceived group variability [87].”

For the “Linking task”, we now write (p. 12):

“The development of the liking task was inspired by the OSIO (Overlap of Self, In-group, and Outgroup) measure [88] and socio-metric methods used to investigate pre-schoolers peer preferences [89]. In our study, children used a game board with five concentric circles (Fig 1) to indicate their liking of refugee peers and German peers (represented by small wooden figures) through spatial proximity on the board.”

8. My understanding is references in brackets should be in alphabetical order (throughout manuscript) and that “and colleagues” requires a citation in brackets to accompany it (e.g., p. 5).

>>>>>> Apologies for the false order of references in brackets in APA 7. As required by PLOS ONE we now have altered the citation style to Vancouver style. References are now numbered consecutively (e.g. pp. 4,5).

We have changed “and colleagues” to “et al.” (p. 5).

Method and Results

9. Table 1: there is no need to repeat the correlation information twice (above and below the table diagonal) (p. 9)

>>>>>> Thank you for spotting this duplication. We have removed the correlation information above the table diagonal in Table 1 (p.9).

10. Could some examples be provided for ‘behavioural examples’ (p. 12)?

>>>>>> Thank you for suggesting sharing the behavioural examples. We have now included them into Table S3 in the S1_File. Please note, however, that we never had the opportunity of using them, as no participant asked for further explanations during testing.

Moreover, we now explain this in the manuscript (p. 11): 

“Instead, we created small drawings (see Figure S1) and decided to only provide behavioural examples (see Table S3) if children failed to understand an adjective. However, no child required further explanation of the adjectives and we thus never used the behavioural examples.”

11. Could more information be provided about the piloting that occurred? (p. 12)

>>>>>> We have now included information on piloting in the “Participants” subsection (p. 9): 

“As we combined different measures in a procedure with young children, we conducted piloting sessions with ten children in an additional kindergarten.”

We moreover included more information on piloting in the task subsections:

Draw-a-typical-child task (p. 12):

“During piloting, both groups of three children and single children performed the draw-a-typical-child task with an experimenter. In the single child setting fewer interruptions occurred and children were able to focus more on the task.”

Intergroup attitude task (p. 11):

“Doyle and Aboud [55] suggested to present the adjectives with behavioural examples, but piloting revealed that children found this confusing and exhausting as it included a lot of verbal information. Instead, we created small drawings (see Figure S1) and decided to only provide behavioural examples (see Table S3) if children failed to understand an adjective. However, no child required further explanation of the adjectives and we thus never used the behavioural examples.”

Liking task (p. 12):

“Piloting revealed children had difficulties understanding the relation between spatial proximity of the figures on the board and liking of the persons they represented. We therefore included a short comprehension task.”

12. “data was” should be “data were” (p. 16 – twice)

>>>>>> Thank you for spotting this! We now write (p. 15):

“but the Bayes Factor BF10 = 1.81, indicated that the data were insensitive” and “but BF10 = 1.75 indicated the data were insensitive”. 

Discussion

13. There is some confusing wording with “not knowing the term “refugee”” (p. 20)

>>>>>> Thank you for pointing to this difficult to understand passage. We now write (p. 19):

“Thus, knowing the term “refugee” may not be necessary for children to show in-group favouritism.”

14. The applied implications referring to encouraging the appreciation of differences etc should be more explicitly linked with the study’s findings. Does it link to the finding about liking German students more than refugee children? (p. 23)

>>>>>> Thank you for these questions and suggestions. We respond to this together with our response to point 15 below.

15. Given the study’s significant findings, it may be beneficial, as part of the applied implications, to encourage thinking about homogeneity/heterogeneity (and that all labelled groups comprise diversity within them) and how that might be translated into intervention strategies? (p. 23)

>>>>>> Thank you for suggesting a clarification of practical implications in our Discussion section (your points 14 and 15). We have now reworded these sections so that they tie in more closely with our findings and theoretical framing

We now write (p. 22):

“Research on children’s socio-cognitive development has overwhelmingly shown that social categorization is a developmental process that allows children to make sense of their social world [30, 37, 111] and forms the basis for perceived group variability and group perceptions [28]. In our study, we found an out-group homogeneity bias towards refugee peers in young children. This may inform pedagogical approaches that encourage children to reflect upon their own and others’ group membership(s) and to think about the variability within social groups (e.g. gender, ethnicity, social origins).”

16. If the Anti-Bias approach is considered feasible and aligned with the study’s findings, can a little more information please be provided about it? (p. 23)

>>>>>> Thank you for raising this point. We have re-examined the Discussion and decided to remove the reference to the Anti-Bias approach, as it does not connect directly with our results.

Recommendation:

The stated points are easily addressed and the manuscript is a very worthwhile contribution to the extant literature. The paper contributes usefully to our understanding of this important and topical area.

>>>>>> Thank you for your very valuable feedback and helpful suggestions. We are glad to hear that you believe that our findings are a worthwhile contribution.

---

## [Editor Report · Decision Letter 1]

8 Jan 2023

Pre-schoolers’ images, intergroup attitudes, and liking of refugee peers in Germany.

PONE-D-22-20046R1

Dear Dr. Würbel,

We’re pleased to inform you that your manuscript has been judged scientifically suitable for publication and will be formally accepted for publication once it meets all outstanding technical requirements.

Kind regards,

Sawsan Abuhammad

Academic Editor

PLOS ONE
---

## [Editor Report · Acceptance letter]

24 Jan 2023

PONE-D-22-20046R1 

Pre-schoolers’ images, intergroup attitudes, and liking of refugee peers in Germany. 

Dear Dr. Würbel:

I'm pleased to inform you that your manuscript has been deemed suitable for publication in PLOS ONE. Congratulations! Your manuscript is now with our production department. 

Kind regards, 

on behalf of

Dr. Sawsan Abuhammad 

Academic Editor

PLOS ONE